# Human Milk Calorie Guide: A Novel Color-Based Tool to Estimate the Calorie Content of Human Milk for Preterm Infants

**DOI:** 10.3390/nu15081866

**Published:** 2023-04-13

**Authors:** Anish Pillai, Susan Albersheim, Nikoo Niknafs, Brian Maugo, Betina Rasmussen, Mei Lam, Gurpreet Grewal, Arianne Albert, Rajavel Elango

**Affiliations:** 1Division of Neonatal-Perinatal Medicine, British Columbia Women’s Hospital and Health Centre, University of British Columbia, Vancouver, BC V6H 3N1, Canada; dranish.pillai@suryahospitals.com (A.P.); susan.albersheim@bcchr.ca (S.A.); brianmaugo@uonbi.ac.ke (B.M.); gurpreet.grewal2@cw.bc.ca (G.G.); 2Department of Neonatology, Surya Hospitals, Mumbai 400054, India; 3British Columbia Children’s Hospital Research Institute, Vancouver, BC V5Z 4H4, Canada; brasmussen@bcchr.ca; 4Department of Pediatrics and Child Health, University of Nairobi, Nairobi 00100, Kenya; 5Women’s Health Research Institute, British Columbia Women’s Hospital and Health Centre, University of British Columbia, Vancouver, BC V6H 3N1, Canada; 6Department of Pediatrics, University of British Columbia, Vancouver, BC V5Z 3V4, Canada

**Keywords:** preterm, growth, human milk, NICU, fortification

## Abstract

Fixed-dose fortification of human milk (HM) is insufficient to meet the nutrient requirements of preterm infants. Commercial human milk analyzers (HMA) to individually fortify HM are unavailable in most centers. We describe the development and validation of a bedside color-based tool called the ‘human milk calorie guide’(HMCG) for differentiating low-calorie HM using commercial HMA as the gold standard. Mothers of preterm babies (birth weight ≤ 1500 g or gestation ≤ 34 weeks) were enrolled. The final color tool had nine color shades arranged as three rows of three shades each (rows A, B, and C). We hypothesized that calorie values for HM samples would increase with increasing ‘yellowness’ predictably from row A to C. One hundred thirty-one mother’s own milk (MOM) and 136 donor human milk (DHM) samples (total *n* = 267) were color matched and analyzed for macronutrients. The HMCG tool performed best in DHM samples for predicting lower calories (<55 kcal/dL) (AUC 0.87 for category A DHM) with modest accuracy for >70 kcal/dL (AUC 0.77 for category C DHM). For MOM, its diagnostic performance was poor. The tool showed good inter-rater reliability (Krippendorff’s alpha = 0.80). The HMCG was reliable in predicting lower calorie ranges for DHM and has the potential for improving donor HM fortification practices.

## 1. Introduction

Human milk (HM) is considered the ideal diet for newborns, including preterm infants [1]. Some of the benefits of HM for preterm infants include a decrease in necrotizing enterocolitis (NEC) rates [2,3], fewer hospital readmissions after discharge from the neonatal intensive care unit (NICU), and improved long-term neurodevelopmental outcomes [4,5]. Preterm infants have higher nutritional requirements and although HM is ideal, unfortified HM is unable to meet their additional nutrient demands [6,7]. Human milk fortifier (HMF) is a multicomponent supplement designed to boost the total energy, protein, and micronutrient content of HM to enhance growth in preterm infants [7,8,9]. The standard fortification policy commonly practiced in NICU’s involves the addition of a fixed-dose HMF to a specified volume of milk. However, this ‘one size fits all’ approach is not ideal because the nutrient content of HM is variable and changes with many factors including the stage of lactation, time of expression, and maternal nutrition [10,11,12,13,14]. Our previous study on corrected fortification showed that standard fortification failed to meet the target macronutrient requirement for preterm infants in more than 90% of HM samples [10]. Thus, preterm infants are at significant risk of extrauterine growth restriction, which has been suggested to impact a multitude of co-morbidities and may impair long-term neurodevelopment [15,16,17].

Alternative approaches to human milk fortification are also practiced. Adjustable fortification involves modification of protein intake based each infant’s metabolic response. Human milk fortification is initiated with a standard multinutrient fortifier until full-strength fortification is tolerated. Then, additional protein fortifiers are added based on the infant’s blood urea nitrogen (BUN) levels, which act as a surrogate for protein adequacy [18,19]. Analysis of HM macronutrient content using the bedside human milk analyzer (HMA) has been validated and is used as a clinical tool to individualize fortification for preterm infants [20,21]. Individualized targeted fortification based on the daily nutrient content of HM has shown encouraging short-term benefits [22,23,24]. However, regular analysis of HM prior to fortification is time-consuming, labor-intensive, and costly. According to a recent American survey of level 3 and 4 NICUs, only 11.8% of centers use HMA technology to guide fortification [25].

Previously published literature [10,11,12] has shown that the protein content of HM consistently decreases with advancing stages of lactation, thus fortification of protein can be predicted and adjusted. However, the fat and calorie content are variable. The data from our previous study on HM fortification showed a huge variation in the calorie content of HM (calorie range 41 kcal/dL to 101 kcal/dL) [10]. We, the research team, and the milk handling room (MHR) staff regularly observed a wide variation in the ‘yellowness’ of the milk samples, with some HM samples having a ‘watery’ grey-blue shade while some appeared dark yellow. We predicted that the yellowness in milk could be primarily attributed to its fat (cream) content and therefore influence calories [26,27,28]. We planned to develop a visual color-based chart as a low-cost, portable tool to predict the calorie (and fat) content of human milk. Our objective in this study was to assess the accuracy of a color-based tool in differentiating low-calorie HM from higher-calorie HM using a commercial HMA (Miris HMA™, Uppsala, Sweden) as the gold standard. The use of color-based tools in clinical practice has been previously described for blood, stool, and urine assessment [29,30,31,32,33]. To the best of our knowledge, such a novel color-based tool has never been described or tested previously for human milk nutrition guidance. The ease of use and practical applicability of such tools could make the human milk calorie guide (HMCG) an attractive option for improving the nutrition and growth of preterm babies.

## 2. Materials and Methods

### 2.1. Study Design and Location 

This prospective study for the development and validation of the HMCG tool was conducted between October 2019 and September 2021 in the British Columbia Women’s Hospital (BCWH) NICU, the milk handling room (MHR), and the British Columbia Children’s Hospital Research Institute (BCCHRI). The NICU is a 60-bed level 3 unit that cares for approximately 700 neonates per year (either inborn or transferred in from other centers in British Columbia (BC) and Yukon). Mothers of preterm babies’ pump breast milk (mother’s own milk (MOM)) in the privacy of their individual rooms in the NICU and store it in the bedside refrigerator. The MHR technicians pick up the milk once daily from the bedside refrigerator to pool, fortify, and dispense the prescribed volume to be fed to each infant. If MOM is insufficient, then pooled donor human milk (DHM) is provided to the preterm infants.

### 2.2. Participants

Mothers of preterm babies with a birth weight ≤ 1500 g or ≤34 weeks gestation were eligible for collection of breastmilk. Collection of MOM for color-matching and macronutrient analysis was done from 2 weeks after delivery up to 8 weeks provided that adequate MOM was available. In addition, individual DHM samples were collected from the BCWH Provincial Milk Bank. The study was reviewed and approved by the Ethics Committee of British Columbia Women’s and Children’s Hospital, Vancouver. Written informed consent to participate in this study was provided by the enrolled mothers of preterm infants.

### 2.3. Development of HMCG Tool

During the conduct of our previously published study on corrected fortification of HM [10], systematic observations by research team members suggested that the yellowness of HM may correlate with its calorie content measured in the laboratory. This inspired the development of a preliminary HMCG tool. After serial inspection of multiple preterm MOM and DHM samples in the MHR by independent observers (neonatal physicians, nurses, MHR technicians, and research assistants), a simple color chart was developed using pictures of HM and commercial paint chip charts [34]. Initially, 20 distinct color shades were selected. This pilot color chart was provided to all MHR staff, and they recorded the most commonly matched colors during their daily milk preparation time over the next three months. If the color of the milk did not match any of the shades on the paint chip chart, it would be noted as such. Six commonly observed shades were then shortlisted. After observing more ‘watery’ gray shades and ‘creamy’ yellow shades from stored MOM and DHM samples, six more shades were added to the chart. The 12 color shades selected were then grouped into 3 rows of 4 colors each. The lighter ‘watery’ shades were grouped in the first row (row A), the ‘regular white’ appearing shades in the second row (row B), and the ‘creamy yellow’ appearing shades were in the third row (row C). The preliminary color chart (preliminary HMCG tool, December 2018) comprised 12 color shades arranged in 3 rows as an equilateral triangle (Figure 1).

Based on preliminary analysis (104 milk samples), certain shades that did not match with sufficient milk samples were eliminated; similarly, shades with poor inter-observer correlation were excluded. Some new shades were added based on direct observations, and a simplified color tool with 9 color shades (3 colors in each row) was created. The final HMCG tool (October 2019 version) is shown in Figure 2.

We postulated that the calorie content of human milk would increase from row A to row B to row C in a predictable fashion. For practical applicability, we predefined the calorie ranges for the color categories on the HMCG tool. We hypothesized that calorie values for milk samples matching with color category A would be <55 kcal/dL, for category B 55–70 kcal/dL, and for category C >70 kcal/dL. The purpose of this tool would be to guide fortification in resource-limited settings where commercial HMA is not readily available.

### 2.4. Milk Collection and Color Coding

For each mother enrolled, two milk samples per week were obtained after the 2nd week. A maximum of 12 samples were provided by each mother. HM was collected and transported to the MHR as per hospital protocol. MOM was aliquoted into 30 mL sterile wide-mouth transparent bottles by the MHR staff. All milk samples were hand-homogenized prior to color matching. Color matching was done by directly observing the milk from the top after opening the lid. The samples were color-coded as per the HMCG tool into one of the 9 shades (Row A: A1, A2, A3; Row B: B1, B2, B3; or Row C: C1, C2, C3). After color matching, milk samples were transported on ice in 3 mL tubes to the laboratory and kept frozen at −20 °C until macronutrient analysis. DHM samples received by the MHR were aliquoted into 30 mL bottles similar to MOM. For tool validation purposes, we used individual DHM samples instead of pooled samples so that we could capture a wider calorie range matching the color categories. The DHM samples were also transported to the laboratory and kept frozen as undertaken with the MOM samples. In the laboratory, batches of DHM samples were thawed to room temperature, and color matching for DHM was performed as with MOM samples. Investigators performing color matching were not aware of HMA results and vice versa. Additionally, for DHM samples, three independent members performed color matching to assess the inter-observer reliability of the tool.

### 2.5. Milk Analysis

The protein, fat, lactose, and calorie content of milk were measured using the commercial mid-infrared HMA. The method used for HM analysis has been previously evaluated in published literature [35,36,37]. A small volume of milk (2 mL) was injected into the HMA for analysis. During the initial development phase of the study, the results from HMA were used to modify the preliminary HMCG tool and assist in deciding the precise calorie ranges that each row/group of shades estimated. During the validation phase, the results from HMA were used to calculate the diagnostic accuracy of the HMCG chart. There was no change in clinical practice, and fortification of feeds for preterm infants in the NICU was continued as per existing unit policy.

### 2.6. Sample Size and Statistical Analysis

For the development of the tool, we planned to collect 40 HM samples for each row or range of calories for a total of 120 samples. This would allow for approximately 10 samples per color shade to help refine the colors and calorie ranges in developing the HMCG tool. For the validation phase, we planned to collect 60 samples per row/group, which would require a minimum of 180 samples. This sample size was estimated based on the variation in human milk fat content and using recommendations on requirements for the minimum sample size for sensitivity and specificity analysis [38]. For the validation phase, we calculated the sensitivity, specificity, and positive and negative predictive value of the HMCG tool as compared to HMA data. We also calculated the accuracy of the tool and performed a ROC curve analysis. The inter-rater reliability between different users of the HMCG tool was measured using Krippendorff’s alpha [39]. To determine if our predetermined calorie cutoff levels were optimal, we used a bootstrapping method; cutpoints were estimated by maximizing Youden’s index using 1000 bootstrap replicates. The returned optimal cutpoint was the mean of the cutpoints across all 1000 replicates. All data analyses were performed using R software V4.1.3 (R Core Team 2022).

## 3. Results

A total of 267 human milk samples (131 MOM and 136 DHM) were color matched and coded using the updated HMCG tool and later analyzed for macronutrient content. Baseline macronutrient data showed that around 22% of DHM samples had calorie content < 55 kcal/dL, and the majority of DHM samples (75%) had calorie levels < 70 kcal/dL. In contrast, most MOM samples (61%) had a calorie content range > 70 kcal/dL (Table 1).

Overall, there was a positive correlation between the yellowness of milk and a higher calorie content for DHM samples (rho = 0.60, *p* < 0.0001 for DHM; rho = 0.11, *p* = 0.22 for MOM; and rho = 0.27, *p* < 0.0001 for combined). For all milk samples combined (MOM plus DHM), the mean calorie content for samples coded as color category A, B, and C were 62.2 kcal/dL, 70.3 kcal/dL, and 73.6 kcal/dL, respectively, as measured by the HMA. However, the subgroup analysis revealed that only the DHM samples had a significant difference in calories across the three color categories. The distribution of calorie ranges for each color category for MOM and DHM is described in Figure 3.

The HMCG tool performed best for predicting lower calories (<55 kcal/dL) in DHM samples (AUC 0.87 for category A DHM) and showed modest accuracy for higher calories (>70 kcal/dL) (AUC 0.77 for category C DHM). For MOM, the diagnostic performance of the tool was poor across all color categories. The overall (MOM plus DHM) accuracy of the HMCG tool was moderate for predicting lower-calorie milk (AUC 0.69 for category A) and poor for predicting higher-calorie content (AUC 0.59 for category C). The diagnostic characteristics of the HMCG tool for DHM and MOM are described in Table 2.

Overall in both MOM and DHM, the majority of milk samples were classified as B category (153 out of 257 samples). Around 37 DHM samples were matched as color category A, and 31 DHM samples were matched as color category C. For MOM, 34 samples were matched as color category A and only 11 samples as category C. The distribution of HM based on assigned color category and measured calorie content is described in Table 3.

Inter-rater reliability was assessed for DHM samples by three independent observers; very good inter-rater reliability was noted at the ABC grouping level (Krippendorff’s alpha = 0.80, 95% CI (0.72–0.87)).

Overall, the optimal cutpoint for category A was estimated to be 57 kcal/dL for MOM and DHM combined and also 57 kcal/dL for donor milk. DHM performed quite well for this cutpoint, and the data suggested that 75% of the samples labeled A were below this threshold (AUC 0.87). The optimal cutpoint for category C was 66 kcal/dL for both MOM and DHM and 67 kcal/dL for DHM, but the accuracy was modest (AUC 0.76). The sensitivity of the HMCG tool for MOM was poor at all cutpoints. The ROC curves for DHM samples in Category A and C are shown in Figure 4.

The distribution of individual macronutrient components showed that fat content significantly increased across three color categories (from A to C, *p* < 0.0001) for DHM but not for MOM samples (*p* = 0.40) (Table 4).

## 4. Discussion

In this study, we described the development and validation of a novel color tool to predict the calorie content (range) of human milk. Our goal was to develop a simple, bedside tool to assist in the fortification of milk for preterm infants in settings where HMAs are not available. In addition, significant variations have been reported in accuracy and precision between different commercial milk analyzers [40]. The HMCG tool showed good accuracy for predicting lower calorie range samples (<55 kcal/dL) for DHM. This will inform better fortification practices and has the potential to improve the growth of premature infants when DHM is the primary source of nutrition.

DHM is used extensively in the NICU for feeding preterm infants, especially in the first few weeks of life when mother’s milk is insufficient [41,42]. We found the calorie content of DHM to be lower than MOM, a finding that has been consistently described in previous studies [43,44,45]. Furthermore, over 20% of DHM samples in our study had very low calorie levels (<55 kcal/dL)—much lower than previous reports. On the other hand, the majority of MOM samples (60%) had calories greater than 70 kcal/dL. The lower calorie content noted in DHM could be related to the collection of individual donor samples rather than pooled samples. Other factors such as the processing of DHM, the stage of lactation of the donor, and the health/nutritional status may impact the total calorie content [43,46].

Nearly two-thirds of our DHM samples would have been under-fortified with the assumption of a fixed calorie content of HM. Evidence is accumulating that early-life growth deficits might have specific impacts on later cognitive and metabolic outcomes [47]. Our practical cost-effective HMCG tool could potentially be very useful and effective in low-resource settings using DHM; recent reports have suggested the need for high-quality research conducted with cost-effective tools [48]. For example, if a preterm baby is on mostly DHM feeds and the color shade of the DHM sample is matched as row A, then the calorie content of that milk is likely to be low (<55 kcal/dL). In such situations, standard fortification is likely to only increase the calorie content of DHM to approximately 68 kcal/dL (additional 13 kcal/dL with 4 sachets HMF). This infant will therefore need an additional calorie intake of 15 kcal/dL from other sources to meet the daily energy requirement.

The use of color-based tools with clinical applicability has been previously described in various healthcare settings. The World Health Organization (WHO) hemoglobin color scale was developed as a simple, inexpensive clinical device for the diagnosis of anemia [32,33]. The utility of stool color charts as a screening tool for cholestasis is well standardized and is currently used as a universal screening strategy in many centers globally [30,31]. Recently, a bilirubin color card tool has been validated for screening neonatal jaundice in low-resource settings [49]. To our knowledge, the use of a color-based tool to estimate the calorie content of human milk has not been described before.

One of the main strengths of the study was the good inter-observer reliability of the HMCG tool for DHM. Many strategies were tested to make the HMCG tool robust and applicable. During the tool-development stage, color matching was performed against various backgrounds (black, white, and stainless-steel countertops). The color matching was performed in different locations by independent observers including the MHR, NICU, and laboratory to ensure it was not affected by changes in room lighting. To study the effect of storage/freezing on milk color, color matching prior to and after freezing and thawing was performed in a subset of samples, which did not show any alteration in color. Other factors that may affect the color reading such as illuminance, the aperture size of the bottle, observer angle, and gentle swirling of the bottle [50] were standardized during the tool-development phase. These findings improved the generalizability of the tool in various clinical settings.

This study had a few limitations. The HMCG tool did not reliably predict the calorie range for MOM samples. As only two MOM samples had calories < 55 kcal/dL, we cannot comment on the accuracy of HMCG in identifying low-calorie MOM. During color matching, only a few HM samples were categorized as C (*n* = 41). Thus, we found limited accuracy in predicting milk samples with higher calories (>70 kcal/dL). We may likely need to develop a separate tool for MOM samples. At this stage, our results might only suggest fortifying the individual low-fat DHM with the HMCG tool. Studies from the dairy industry have shown that the yellowness of milk may not only be related to the fat content. One factor that may impact the yellowness of milk is the presence of carotenoids [51]. However, major carotenoids present in HM are in several fractions, including β-carotene, lycopene, lutein, β-cryptoxanthin, zeaxanthin, and α-carotene. With the limited volume of milk collected, we were unable to measure the carotenoids with reliable precision in our core laboratories. The impact of diet on the color of cow’s milk has been reported [26]. We did not record diet history for mothers or donors, and this will be an additional measure in the future along with the measurement of milk carotenoids. Finally, we used the mid-infrared HMA as the gold standard, which may slightly underestimate the protein content and overestimate the carbohydrate content [52].

## 5. Conclusions

The present study described the development and validation of a color-based tool—HMCG—as a screening tool to predict HM calorie range. The tool was most reliable in predicting lower calorie ranges for DHM samples and will be useful in improving donor HM fortification practice. While larger studies are required in this area to confirm our results and improve the precision of our tool, the impact of this tool on growth outcomes in preterm infants needs to be explored in future clinical trials. The HMCG color-based tool has a great potential for practical application in both resource-limited settings and NICUs with established MHR where DHM is predominantly used and when commercial HMAs are not available.

## Figures and Tables

**Figure 1 nutrients-15-01866-f001:**
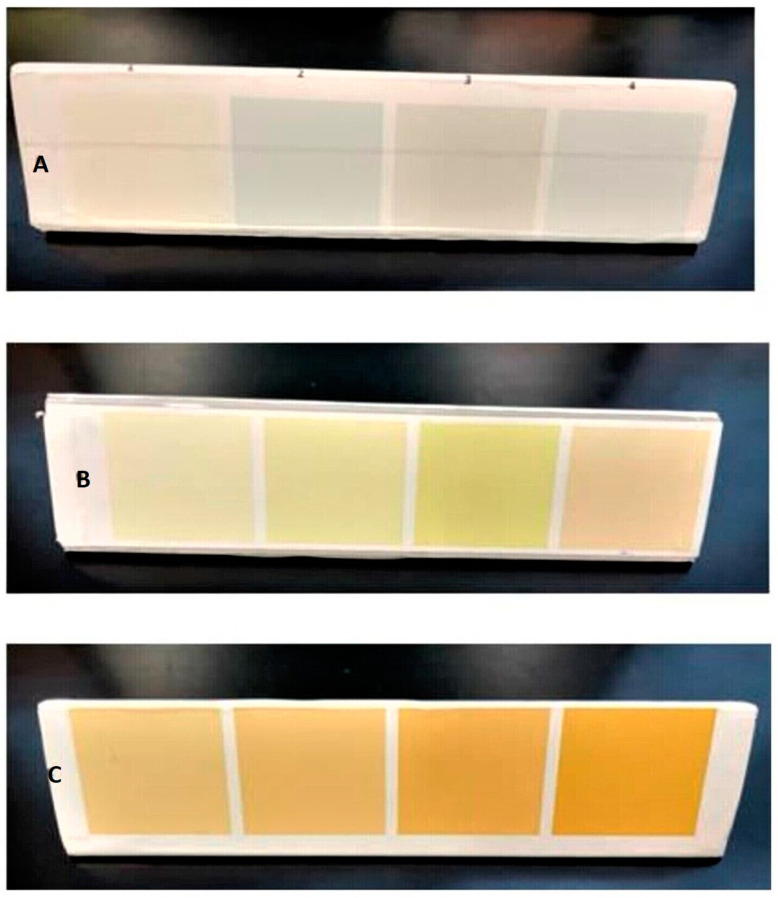
Preliminary HMCG tool comprising 12 shades (December 2018 version). The lighter ‘watery’ shades are grouped in the first row (row **A**), the ‘regular white’ appearing shades in the second row (row **B**), and the ‘creamy yellow’ appearing shades are in the third row (row **C**).

**Figure 2 nutrients-15-01866-f002:**
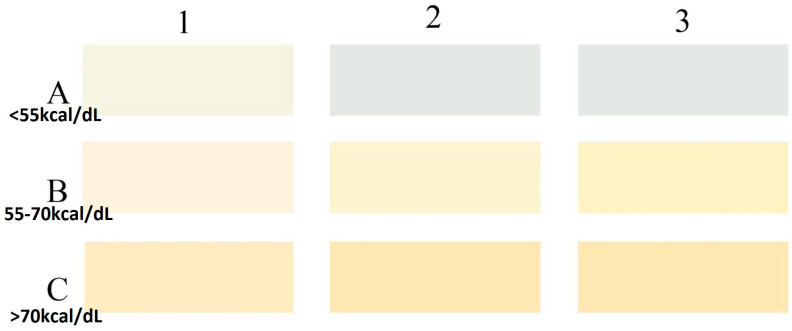
HCMG tool with 9 color shades (October 2019 version). Note: colors from www.Behr.com [35]. A1—Cauliflower M340-1 White, A2—Silent White PPU26-13, A3—Wind Chill BL-W07; B1—Glass of Milk P260-1, B2—Effervescent P310-1, B3—Natural Light P310-2; C1—Yogurt P260-2, C2—Vanilla Ice Cream P260-3, C3—Hummus PPU6-11.

**Figure 3 nutrients-15-01866-f003:**
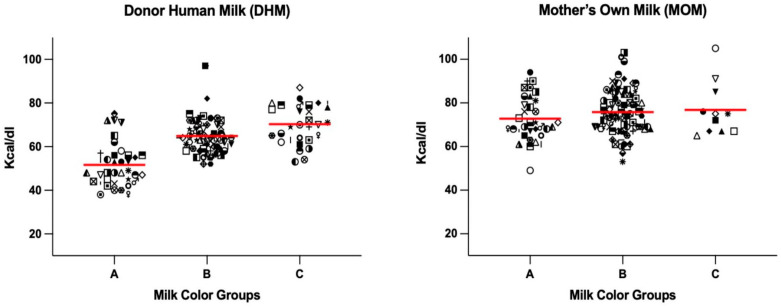
Average calories with data scatter in each color category. The *x*-axis represents the milk samples color-matched and coded as A, B, or C categories. The *y*-axis represents the calorie value of human milk as measured by the human milk analyzer. DHM A, B, and C, category mean calories = 52, 65, and 70 kcal/100 mL, respectively; MOM A, B, and C category mean calories = 73, 76, and 77 kcal/dL, respectively.

**Figure 4 nutrients-15-01866-f004:**
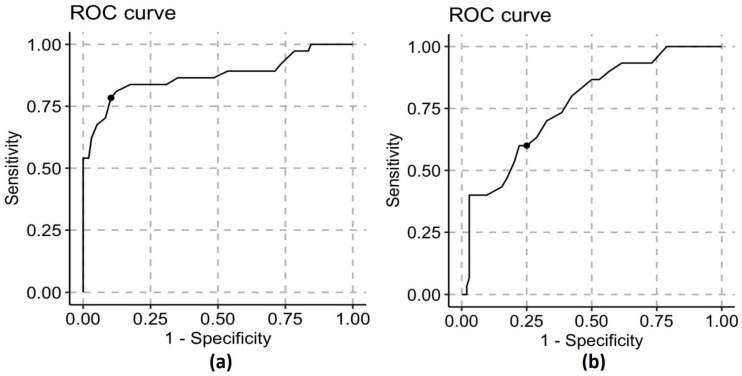
ROC curve analysis for DHM samples in category A (**a**) and category C (**b**). The AUC for Category A was 0.87; for Category C, it was 0.76.

**Table 1 nutrients-15-01866-t001:** Distribution of human milk samples as low, expected, or high-calorie ranges.

Calorie Range (kcal/100 mL)	DHM (*n* = 136)	MOM (*n* = 131)
<55 (*n*, %)	30 (22.1%)	2 (1.5%)
55–70 (*n*, %)	72 (52.9%)	49 (37.1%)
>70 (*n*, %)	34 (25.0%)	80 (60.6%)

DHM, donor human milk; MOM, mother’s own milk.

**Table 2 nutrients-15-01866-t002:** Diagnostic ability of HMCG tool for predicting HM calorie ranges.

DHM	A (<55 kcal/dL)	B (55–70 kcal/dL)	C (>70 kcal/dL)
Sensitivity	0.83	0.69	0.44
Specificity	0.84	0.70	0.82
PPV	0.68	0.75	0.47
NPV	0.93	0.64	0.81
**MOM**	**A**	**B**	**C**
Sensitivity	0.50	0.59	0.09
Specificity	0.52	0.12	0.88
PPV	0.03	0.34	0.64
NPV	0.97	0.29	0.29

DHM, donor human milk; MOM, mother’s own milk; PPV, positive predictive value; NPV, negative predictive value.

**Table 3 nutrients-15-01866-t003:** Classification of milk samples based on calorie content and assigned color category.

DHM	A	B	C
<55 (kcal/100 mL)	25 (67.6%)	3 (4.4%)	2 (6.4%)
55–70 (kcal/100 mL)	8 (21.6%)	50 (73.5%)	14 (45.1%)
>70 (kcal/100 mL)	4 (10.8%)	15 (22.0%)	15 (48.3%)
**MOM**	**A**	**B**	**C**
<55 (kcal/100 mL)	1 (2.9%)	1 (1.1%)	0 (0.0%)
55–70 (kcal/100 mL)	16 (47.1%)	29 (33.3%)	4 (36.4%)
>70 (kcal/100 mL)	17 (50.0%)	56 (64.4%)	7 (63.6%)

DHM, donor human milk; MOM, mother’s own milk.

**Table 4 nutrients-15-01866-t004:** Distribution of macronutrients across color categories given as the mean (SD).

DHM	A	B	C	*p* Value
Fat (g/100 mL)	1.6 (0.8) ^a^	3.4 (1.0) ^b^	3.9 (1.0) ^c^	<0.0001
Protein (g/100 mL)	1.1 (0.1) ^a^	1.1 (0.3) ^a^	1.4 (0.5) ^b^	0.0001
Carbohydrate (g/100 mL)	7.3 (0.3) ^a^	7.2 (0.4) ^a^	6.9 (0.4) ^b^	0.002
**MOM**	**A**	**B**	**C**	
Fat (g/100 mL)	4.2 (1.2)	4.5 (1.1)	4.6 (1.4)	0.40
Protein (g/100 mL)	0.8 (0.1)	0.9 (0.3)	1.0 (0.2)	0.20
Carbohydrate (g/100 mL)	7.2 (0.2)	7.2 (0.2)	7.1 (0.2)	0.26

Note: *p*-values are for the overall comparison from an ANOVA. Pairwise differences are indicated as superscript letters. If they have the same letter, then they were not different at *p* < 0.05. DHM, donor human milk; MOM, mother’s own milk.

## Data Availability

The raw data supporting the conclusions of this article will be made available by the authors without undue reservation.

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
