# Peer review of "Human Milk Calorie Guide: A Novel Color-Based Tool to Estimate the Calorie Content of Human Milk for Preterm Infants"

_nutrients, 2023, doi:10.3390/nu15081866_

Round 1
Reviewer 1 Report
Manuscript Nutrients-2313598
The manuscript entitled “Human Milk Calorie Guide: A novel color-based tool to estimate calorie content of human milk” by Pillai, A. et al. presents a bedside color-based tool to estimate calories in human milk samples. The manuscript is well-written and organized, and although the performance of the approach is not excellent, it deserves especial attention in clinical practice. In my opinion, the manuscript is suitable for publication under minor modifications.
· In the introduction section, the authors state that standard fortification is the commonly practice in NICU’s and it is compared with targeted fortification. However, other fortification strategies such as adjustable fortification guided by blood urea nitrogen should be mentioned.
· Figure 1. Please, include A, B and C labels to the different rows since in the picture cannot be appreciated.
· Figure 2. Please, for clarity, include de calorie ranges (i.e., < 55 kcal/dl, 55-70 kcal/dl, >70 kcal/dl) in the A, B and C rows, respectively.
· Revise abbreviations (e.g., line 133 human milk analyzer)
· Table 1. Include in the table caption or in the first column that the distribution of samples was made according to color-match.
· Lines 188-193. I assume that these results correspond to the HMA. Please, indicate.
· Figure 3. In the figure caption, please, specify that average categories correspond to the color-match approach and that y-axis correspond to results provided with the HMA.
· Table 3. Revise numbers. The sum of DHM samples is 134 not 136 as described along the manuscript. Regarding table caption, instead of ‘Distribution’, the term ‘classification’ seems more appropriate to me.
· Figure 4. A and B labels are not included in the Figure and in the table caption they are referred as Figure3A and Figure3B.
Author Response
Response to Reviewer 1 comments:
We thank the reviewer for comprehensively reviewing our manuscript and providing a detailed and constructive feedback. We have replied to all suggestions in a structured, point-wise manner.
- a) In the introduction section, the authors state that standard fortification is the commonly practice in NICU’s and it is compared with targeted fortification. However, other fortification strategies such as adjustable fortification guided by blood urea nitrogen should be mentioned.
Thank you for the suggestion. We have expanded the section of fortification practices, mentioning adjustable fortification as well (line 55-59)
- b) Figure 1. Please, include A, B and C labels to the different rows since in the picture cannot be appreciated.
We have edited Figure 1 and added clear labels A, B and C for the rows.
- c) Figure 2. Please, for clarity, include de calorie ranges (i.e., < 55 kcal/dl, 55-70 kcal/dl, >70 kcal/dl) in the A, B and C rows, respectively.
We have included the calorie ranges as suggested by you in Figure 2.
- d) Revise abbreviations (e.g., line 133 human milk analyser)
We have revised the abbreviation and edited the sentence
- e) Table 1. Include in the table caption or in the first column that the distribution of samples was made according to color-match.
We would like to clarify that in table 1, the distribution of samples is described as per the calorie range (measured by HMA) and not as per color-matching. The distribution of human milk samples as per color-match is described later in Table 4. Hence, we have not edited this table caption.
- f) Lines 188-193. I assume that these results correspond to the HMA. Please, indicate.
We have added the statement to clarify that the milk calorie levels were measured and reported as per HMA results.
- g) Figure 3. In the figure caption, please, specify that average categories correspond to the color-match approach and that y-axis correspond to results provided with the HMA.
We have edited the figure legend as per your suggestion to clarify the descriptions of the x-axis and y-axis.
- h) Table 3. Revise numbers. The sum of DHM samples is 134 not 136 as described along the manuscript. Regarding table caption, instead of ‘Distribution’, the term ‘classification’ seems more appropriate to me.
We thank you for bringing this to our attention. We have revised the numbers after verifying our raw data excel-sheet. We have also modified the caption of Table 4 as per your suggestion.
- i) Figure 4. A and B labels are not included in the Figure and in the table caption they are referred as Figure3A and Figure3B.
We have added the labels Figure 4a and Figure 4b to the original figure, as mentioned in the caption.
Reviewer 2 Report
Dear Authors,
Congratulations on submitting a very interesting study on human milk calorie estimation.
Title
The title of the manuscript draws attention of the reader. It clearly defines the which tool was evaluated; however, it does not define the study population. I feel that the population of preterm infants should be mentioned as this clarifies, who may be the potential beneficent.
Abstract
The authors provide a concise overview of the study. It is clearly stated why the study will be undertaken, and how the data was collected.
Introduction
The authors provide enough contextual information for the journal’s neonatal-paediatric readership to understand the context. They accurately describe up to date research, which is properly referenced. Additionally, limitations and controversies are identified.
Methods
Methods and statistical analysis are adequately planned. However, I feel that additional information on milk collection methods should be included, for example did Mothers performed 24-hour milk collection? Which week of breastfeeding where the samples collected?
Sample size is adequate, and the outcomes are clearly defined. The statistics are sound, and the study was conducted under ethical guidance.
Discussion
The discussion addresses the research problem. Other relevant studies are discussed.
However, given how standard human fortifiers are administered I would like to ask the authors how this notably interesting tool can be put into practice? i.e. how will the clinical decide on how much additional fortifier does the baby require, and which milk content?
I look forward to reading the revised manuscript!
Author Response
We thank the reviewer for comprehensively reviewing our manuscript and providing a detailed and constructive feedback. We have replied to all suggestions in a structured, point-wise manner.
Title
The title of the manuscript draws attention of the reader. It clearly defines the which tool was evaluated; however, it does not define the study population. I feel that the population of preterm infants should be mentioned as this clarifies, who may be the potential beneficent.
Thank you for your suggestion. We have modified our title to include the population for which the tool would be most beneficial, i.e preterm infants. We suggest the following title:
“Human Milk Calorie Guide: A novel color-based tool to estimate the calorie content of human milk for preterm infants”
Abstract
The authors provide a concise overview of the study. It is clearly stated why the study will be undertaken, and how the data was collected.
Thank you for the positive feedback
Introduction
The authors provide enough contextual information for the journal’s neonatal-paediatric readership to understand the context. They accurately describe up to date research, which is properly referenced. Additionally, limitations and controversies are identified.
Thank you for your encouraging words
Methods
Methods and statistical analysis are adequately planned. However, I feel that additional information on milk collection methods should be included, for example did Mothers performed 24-hour milk collection? Which week of breastfeeding where the samples collected?
We have added some additional details regarding milk collection and analysis in the methods section.
Sample size is adequate, and the outcomes are clearly defined. The statistics are sound, and the study was conducted under ethical guidance.
Thank you.
Discussion
The discussion addresses the research problem. Other relevant studies are discussed.
However, given how standard human fortifiers are administered I would like to ask the authors how this notably interesting tool can be put into practice? i.e. how will the clinical decide on how much additional fortifier does the baby require, and which milk content?
We have incorporated additional details regarding how our color-tool could improve human milk fortification practices in the NICU as per your suggestion. Lines 311-317